# Structural Basis of Arrestin Selectivity for Active Phosphorylated G Protein-Coupled Receptors

**DOI:** 10.3390/ijms222212481

**Published:** 2021-11-19

**Authors:** Preethi C. Karnam, Sergey A. Vishnivetskiy, Vsevolod V. Gurevich

**Affiliations:** Department of Pharmacology, Vanderbilt University, Nashville, TN 37232, USA; preethi.c.karnam@vanderbilt.edu (P.C.K.); sergey.vishnivetskiy@vanderbilt.edu (S.A.V.)

**Keywords:** arrestin, GPCR, phosphorylation, selectivity, structure–function

## Abstract

Arrestins are a small family of proteins that bind G protein-coupled receptors (GPCRs). Arrestin binds to active phosphorylated GPCRs with higher affinity than to all other functional forms of the receptor, including inactive phosphorylated and active unphosphorylated. The selectivity of arrestins suggests that they must have two sensors, which detect receptor-attached phosphates and the active receptor conformation independently. Simultaneous engagement of both sensors enables arrestin transition into a high-affinity receptor-binding state. This transition involves a global conformational rearrangement that brings additional elements of the arrestin molecule, including the middle loop, in contact with a GPCR, thereby stabilizing the complex. Here, we review structural and mutagenesis data that identify these two sensors and additional receptor-binding elements within the arrestin molecule. While most data were obtained with the arrestin-1-rhodopsin pair, the evidence suggests that all arrestins use similar mechanisms to achieve preferential binding to active phosphorylated GPCRs.

## 1. Introduction

G-protein coupled receptors (GPCRs) are the largest known family of signaling proteins, with over 800 members in humans, and even more in most mammalian species (http://gpcrdb.org, accessed on 16 on November 2021). They are responsible for initiating intracellular signaling that affects metabolism, growth, differentiation, and mediate sensory inputs underlying taste, sense of smell, and vision [1]. GPCRs are targeted by about a third of clinically used drugs [2]. While there is a lot of structural and dynamic information about how GPCRs engage G-proteins, less is known about how arrestins interact with GPCRs. Arrestins are regulatory proteins that play a key role in homologous desensitization of GPCRs (Figure 1) and direct signaling to other pathways [3,4]. Existing structures of two different arrestins (arrestin-1 and arrestin-2) bound to several GPCRs [5,6,7,8,9,10] reveal the end result, but do not reveal the sequence of molecular events during binding or identify arrestin and receptor elements that drive the process. Receptor binding of arrestins is precisely timed, and arrestins demonstrate impressive selectivity for active phosphorylated receptors [11,12]. Timing and selectivity of arrestin binding was shown to be critical for the proper function of rod photoreceptors in the retina [13], as well as in other GPCR-driven signaling systems [14].

Only four subtypes of arrestin proteins are expressed in most vertebrates (bony fish that underwent an extra round of whole genome duplication express more) [15]. Arrestin-1 and -4 are the two visual subtypes present in photoreceptor cells in the retina. Their function is to regulate phototransduction cascades, i.e., quench photopigment signaling in rods and cones [16]. Arrestin-2 and -3 are ubiquitously expressed and regulate hundreds of GPCR subtypes involved in a wide range of signaling pathways [3,4]. Arrestin-1 displays the highest selectivity for active phosphorylated rhodopsin, as compared to its other functional forms [11], including retinal-free opsin [17]. Therefore, it was used to elucidate the mechanism of arrestin activation and its binding to various functional forms of cognate receptors. Similar to rhodopsin, non-visual GPCRs can be in an active or inactive state, unphosphorylated or phosphorylated. It should be noted, though, that in contrast to rhodopsin, which exists in the dark with covalently attached inverse agonist 11-cis-retinal, effectively suppressing its constitutive activity (reviewed in [18]), non-visual GPCRs in an unliganded state, as well as occupied by various ligands, sample a variety of conformations [19]. Additionally, in rhodopsin, all GRK-phosphorylated residues are compactly localized in a short stretch in the C-terminus [13], while in other receptors, GRK targets can be in the C-terminus or cytoplasmic loops, often in more than one receptor element (e.g., see several examples in [20] and more general discussion in [3]). However, critical mechanisms that govern the arrestin–GPCR interactions appear to be conserved in other arrestin subtypes [11], although their relative roles in the binding vary. Two possible modes of the binding of non-visual arrestins to their cognate GPCRs have been documented: the engagement of only the phosphorylated C-terminus and simultaneous two-site interaction with the C-terminus and the cavity between transmembrane helices [21]. Interestingly, the first mode of binding was shown to be compatible with simultaneous interaction with arrestin and G protein [22]. Even though this was demonstrated using an artificial construct, β2-adrenergic receptor (β2AR) with vasopressin V2 receptor C-terminus, it is entirely possible that this might be applicable to natural non-visual GPCRs. Arrestin-1 demonstrates virtually no binding to unphosphorylated inactive rhodopsin (Rh) or unphosphorylated opsin [17]. Relatively low binding to unphosphorylated, active rhodopsin (Rh*), phosphorylated inactive rhodopsin (P-Rh), and phosphorylated opsin was detected [17]. In fact, phosphorylated opsin is the second most “attractive” functional form of rhodopsin after phosphorylated active rhodopsin (P-Rh*) [17,23]. Arrestin-1 demonstrates 10- to 20-fold higher binding to P-Rh* [17].

This high specificity was explained by the existence of two independent elements (termed sensors [17]) in arrestin-1, one interacting with the receptor-attached phosphates and the other with parts of the receptor that change conformation upon activation [11,17]. The sequential multi-site model of the arrestin–receptor interaction posits that to swing into action, arrestins need both the receptor in active conformation and receptor-attached phosphates to simultaneously engage arrestin elements that specifically bind respective parts of the receptor [11]. In the case of arrestin-1, the only functional form of rhodopsin capable of engaging both sensors at the same time is P-Rh* [17]. This triggers a global conformational change in arrestin, which brings to the receptor additional elements that stabilize this interaction, and therefore increase the binding affinity [11]. This model is consistent with available structures of free (in the basal conformation) [24,25,26,27] and receptor-bound [5,6,7,8,9,10] arrestins, and molecular modeling of arrestin activation [28]. Conformational changes in arrestin-3 (i.e., β−arrestin2) induced by GPCR binding were explored in cells using several sensors incorporated into the arrestin molecule [29,30]. Additionally, differentially phosphorylated rhodopsin [31] and vasopressin receptor C-termini [32] were used to investigate how the pattern of receptor phosphorylation (usually termed “barcode”) affects arrestin binding and binding-induced conformational rearrangements. Nonetheless, the exact mechanism of the conformational change in arrestin induced by GPCR binding has not been fully elucidated. The first step in this direction is the identification of arrestin sensors. 

## 2. Phosphate Sensor

Several hypotheses were proposed regarding the identity of arrestin elements that serve as the phosphate sensor that interact with the phosphorylated C-terminus of rhodopsin [33,34,35] and as activation sensor [36,37,38]. Phosphate binding and the identity of the phosphate sensor have been extensively investigated. Three candidates conserved in all vertebrate arrestin subtypes [15] were explored in depth: the polar core arginine [35], lariat loop lysine [33], and two lysines in the β-strand I [34] (Figure 2 and Figure 3).

Early mutagenesis identified several important functional regions in the arrestin molecule, including the N-terminal domain (residues 1–191) of arrestin-1, with elements implicated in both activation recognition and phosphorylation recognition [17]. In visual arrestin-1, when P-Rh* binds and triggers arrestin-1 transition into a high-affinity receptor-binding state, the hypothesized global conformational rearrangement was shown to mobilize an additional hydrophobic binding site(s) [17]. The engagement of secondary binding site(s) greatly increases the affinity for P-Rh*, which explains visual arrestin-1 selectivity for this form of rhodopsin. Before any structures were solved, the intramolecular interaction of the N- and C-termini of arrestin-1 was shown to regulate the conformational rearrangement necessary for the mobilization of secondary binding site(s) [39]. The deletion of either the N- or C-terminal elements yields mutant arrestins that bind readily not only to P-Rh*, but also to inactive P-Rh and Rh* [39]. Deleting the whole C-terminal domain yields an arrestin-(1–191) that is capable of activation and phosphorylation recognition via the primary binding sites but has lower selectivity to P-Rh* compared to WT, apparently due to a loss of the secondary site(s) [17]. The phosphate-induced increase of arrestin binding to the receptor requires that the phosphorylated C-terminus of rhodopsin destabilizes some of the interactions that constrain free arrestin in the basal state (here, by basal, we mean the conformation of free arrestins, which is remarkably similar in the crystal structures of all subtypes [24,25,26,27,40,41,42]). The phosphates are negatively charged at physiological pH, which suggests that the negatively charged phosphorylated C-terminus of rhodopsin likely interacts with positively charged residues in arrestin [43].

Early mutagenesis studies identified several bovine arrestin-1 residues, including Arg175, that directly interact with the receptor-attached phosphates [43]. Several point mutations were introduced on the wild-type (WT) background in the region including residues 163–191 containing nine positively charged residues, four of which are conserved in the non-visual homologs [43]. To specifically address the question of the importance of charge, these positively charged residues were substituted with uncharged residues capable of hydrogen bonding: lysines were replaced by serines, arginines, and histidine by glutamines or asparagines [43]. On the WT background, the Arg175Asn mutation increased arrestin-1 binding to Rh* dramatically, virtually to the level of that of WT arrestin-1 binding to P-Rh*. This is consistent with the role of Arg175 as the phosphate sensor in arrestin-1 [11]. Since binding of full-length arrestin is dependent on its ability to interact with rhodopsin’s phosphorylated C terminus and to mobilize secondary binding site(s), the same mutations were also introduced into “mini-arrestin” (1–191), which does not have any additional receptor-binding sites. On this background substitution of Arg175 reduced arrestin-1 binding to both inactive P-Rh and P-Rh* [43], suggesting that facilitation of the engagement of secondary sites was responsible for the positive effect on the full-length background [43]. The role of Arg175 in P-Rh* binding was later confirmed by another lab using a different method [44]. 

### 2.1. Polar Core

The cystal structure of the basal arrestin-1 [24,40], which was solved later, revealed two key intramolecular interactions that hold the molecule in its basal state, which is often considered inactive. These include: 1) the three-element interaction between the arrestin C-tail, β-stand I, and α-helix I; and 2) the polar core, a network of five charged residues between the two domains (Figure 2). Disruption of either of these interactions results in enhanced binding to inactive P-Rh, Rh*, and P-Rh* [45], suggesting that at least one of these interactions is responsible for phosphate sensing (reviewed in [11]). Within the polar core, there are three negatively charged aspartic acids and two positively charged arginines (Figure 2) in a delicate charge balance. It was shown that in bovine arrestin-1, the critical polar core interaction is between Arg175 and Asp296 [35]. Charge reversal of either residue greatly increased Rh* binding, as if the phosphate sensor was turned “on” by these mutations, so that arrestin-1 “perceived” any activated rhodopsin as phosphorylated [35]. Simultaneous reversal of both charges restored the salt bridge and arrestin-1 selectivity for P-Rh*. Based on these data, it was suggested that receptor-attached phosphates bind the polar core arginine and neutralize its charge, disrupting the salt bridge between Arg175 and Asp296, and that this disruption serves as the signal for arrestin-1 that the phosphates are in place [11]. It should be noted that comprehensive alanine-scanning arrestin-1 mutagenesis [44] confirmed the critical role of the polar core charges in arrestin-1 binding to rhodopsin. Charge elimination of Asp 30, Arg175, Asp296, Asp303, and Arg382 increased arestin-1 binding to P-Rh* determined by a different method [44]

Homologous to Arg-175 residues Arg-169 in bovine arrestin-2 and Arg-170 in bovine arrestin-3 are located in similar polar cores of these proteins [25,27]. Similar to arrestin-1, charge reversal of these residues allowed binding to non-phosphorylated active forms of their cognate non-visual GPCRs [46,47,48,49].

Numerous mutants of arrestin-1, -2, and -3 have been constructed and tested over the years, some of which demonstrated increased binding to P-Rh* and even unphosphorylated Rh*. When tested, these mutants demonstrate greater overall flexibility [50] and greater propensity to spontaneously assume a receptor-bound-like conformation [51]. These mutants in different studies were termed “enhanced”, “pre-activated”, or “active”. Below, we use the term “pre-activated”, indicating their increased binding to cognate receptors. We do not believe the term “active” correctly describes these mutants, as pre-activated mutants of non-visual arrestins do not facilitate ERK1/2 activation without binding to receptors [52,53], indicating that the conformations of even the most pre-activated mutants are distinct from those of receptor-bound arrestins.

While mutagenesis studies provided a compelling case for the polar core arginine serving as the phosphate sensor, the crystal structures of the arrestin-receptor complexes did not support this idea: this arginine was not in contact with receptor-attached phosphates in any of the structures of the arrestin-receptor complex [5,6,7,8,9,10], or with any of the IP6 phosphates in the structure of arrestin-3 trimer in complex with IP6, where each protomer was in a receptor bound-like conformation [38]. The structures of three arrestins bound to different receptors or phosphorylated GPCR elements were published (Table 1): arrestin-1 in complex with phosphorylated rhodopsin [5,6], arrestin-2 in complex with neurotensin [10] receptor, pre-activated human arrestin-2-Arg169Glu mutant in complex with β1-adrenergic receptor [9], truncated arrestin-2 (without the C-terminus) in complex with neurotensin [7] and M2 muscarinic cholinergic [8] receptors; arrestin-2 in complex with multi-phosphorylated C-terminal peptide of human V2 vasopressin receptor [54]; and arrestin-3 in complex with phosphorylated CXCR7 C-terminal peptide [55]. The polar core arginine did not interact with phosphates in any of these structures. In contrast, a conserved lysine in the lariat loop of all arrestins was found to be in contact with one of the activator-attached phosphates (Figure 2).

### 2.2. Lariat Loop Lysine

Therefore, an alternative mechanism, whereby phosphates can destabilize the polar core and induce arrestin activation, was proposed [37]. Lysine in the lariat loop is sandwiched between the two residues providing negative charges to the polar core. Thus, it was proposed that activator-attached phosphates bind this lysine and pull the lariat loop out of its basal position, which would destabilize the polar core by the removal of the two negative charges, triggering arrestin activation [37].

The role of the lariat loop lysine was tested experimentally by mutagenesis [33]. Charge reversal and neutralization mutants (Lys->Glu and Lys->Ala, respectively) were introduced in bovine and mouse arrestin-1, as well as in bovine arrestin-2 and -3. These mutations were introduced on the WT background, where they were expected to reduce the binding to phosphorylated receptors, and on the background of several structurally distinct pre-activated mutants, where substitutions of this lysine were expected to affect the binding a lot less [33]. The four mutations known to “pre-activate” arrestin-1, greatly increasing the binding to unphosphorylated Rh*, were used: triple alanine substitution of bulky hydrophobic residues that anchor the C-tail to the N-domain (3A mutation) [45] (Figure 2), the C-tail deletion [45], and individual charge reversals of the two critical residues that form the key salt bridge in the polar core [35].

It turned out that lariat loop lysine mutations do not have a significant effect on the binding of WT arrestins to P-Rh* and phosphorylated non-visual receptors [33]. In fact, charge reversal and neutralization of the lariat loop lysine on some pre-activated backgrounds in arrestin-1 even increased the binding to Rh*, suggesting that it might be involved in interactions with unphosphorylated portions of rhodopsin. Thus, while the conserved lysine in the lariat loop apparently participates in receptor binding, it does not act as the phosphate sensor [33]. This lysine is not the only lariat loop residue implicated in rhodopsin binding: alanine substitutions of Leu249, Tyr250, Ser251, Ser252, and Tyr254 in the lariat loop were shown to reduce P-Rh* binding [44].

### 2.3. Two Lysines in the β-Strand I

This left only one possible candidate for the role of the phosphate sensor: two lysines in the β-strand I of all arrestins [11] (Figure 3). The N-domain of arrestins has multiple exposed positive charges, including the two lysines in β-strand I (Lys-14, -15 in bovine arrestin-1, homologous Lys-10, -11 in bovine arrestin-2, and Lys-11, -12 in bovine arrestin-3). The replacement of Lys-14 and -15 with alanines was shown to dramatically reduce P-Rh* binding of WT bovine arrestin-1 but not the binding of pre-activated mutants, where the basal conformation was destabilized by various mutations [34]. Alanine substitution of these two lysines in arrestin-1 virtually abolished binding to P-Rh*. Most strikingly, Lys15Ala mutation alone reduced the binding by ~80% [34], making it the most detrimental point mutation described so far. Lys14Ala and Lys15Ala mutations also eliminated arrestin interaction with inactive P-Rh, which is driven solely by phosphate binding [17,34]. These data suggested that one or both conserved lysines in the β-strand I likely serve as the phosphate sensor and their engagement by the phosphate triggers activation-associated conformational changes in arrestin. Importantly, the structures of the arrestin-receptor complexes, as well as the structure of arrestin-3 in receptor bound-like conformation in complex with IP6, revealed that the first of the two lysines in β-strand I (Figure 3) consistently interacts with one of the phosphates of the activator [6,7,8,38,54].

Collectively, the available evidence identifies the two lysines in the β-strand I as the key phosphate-sensing residues [33]. In the basal state, the side chains of these adjacent lysines face in opposite directions (Figure 3), so for both lysines to contact the phosphate, one of them must flip. This conformational rearrangement would destabilize the short β-strand I. As a result, adjacent large hydrophobic residues in β-strand I that interact with their counterparts in the arrestin C-tail and α-helix I would move out of the positions favorable for these interactions, thereby disrupting the three-element interaction that stabilizes the basal arrestin conformation (Figure 2). C-tail release removes another arginine (Arg-382 in bovine arrestin-1 and its homologues in other arrestins (Figure 2)) from the polar core, thereby relieving all constraints that prevent arrestin from achieving the high-affinity receptor-binding state [11]. The destabilization of the three-element interaction and consequent release of the C-tail was shown to enhance the binding of arrestin-1, -2, and -3 to their cognate receptors by several labs using a variety of methods [23,33,44,45,46,48].

The role of the two lysines in the β-strand I in binding to cognate GPCRs of non-visual arrestin-2 and -3 was tested using the in-cell BRET-based assay [56,57]. The binding of both non-visual subtypes to P-Rh* in vitro and the binding of arrestin-1 to non-visual GPCRs in cells was dramatically reduced by alanine substitutions of these two lysines [57]. These interactions with non-cognate receptors were likely largely driven by the phosphate binding, so that these data support the idea that these two lysines are critical for phosphate-induced arrestin activation. However, the binding of both arrestin-2 and -3 to cognate non-visual GPCRs in cells was not dramatically impacted by the elimination of positive charges in these positions [56,57]. Thus, non-visual arrestins do not appear to rely as much on receptor-attached phosphates as arrestin-1.

Both receptor phosphorylation and activation are needed for high-affinity arrestin binding, although non-visual arrestin-2/3 are less selective than arrestin-1 (Figure 4). Both sensors must be engaged simultaneously to trigger arrestin transition into a high-affinity receptor-binding conformation, as evidenced by direct binding data [17,47,49,58] and suggested by modeling [28]. When arrestin interacts with the active phosphorylated receptor, the phosphate sensor engages receptor-attached phosphates, while the activation sensor binds parts of the receptor that change conformation upon activation, destabilizing inter-domain interactions. This model of arrestin binding suggests functional symmetry between phosphorylation and activation (reviewed in [11]). While the mechanism of phosphate sensing was thoroughly investigated, the identity of the activation sensor is still debated.

## 3. Finger Loop as an Activation Sensor

Arrestin finger loop connecting β-strands V and VI emerged as the top candidate for the receptor activation sensor in visual arrestin-1 [36] (Figure 5). The sequence of this loop is remarkably conserved among arrestin subtypes, and the finger loop was invariably found in the cavity between the cytoplasmic ends of GPCR transmembrane helices in all structures of the complex [5,6,7,8,9,10]. As GPCR activation is always accompanied by the opening of this cavity [19,59], it makes perfect sense that the presence of an open cavity serves as an indicator of receptor activation for all proteins that prefer active GPCRs over inactive, G proteins, G protein-coupled receptor kinases, and arrestins [12,60,61]. In the protomers where it is resolved, the finger loop in the basal state of arrestin-1 is unstructured [24,40,42]. This loop forms a short α-helix upon binding rhodopsin [5,6] (Figure 5). While a similar conformational change in the finger loop appears to occur in arrestin-2 bound to the neurotensin receptor 1 [7,10], it was not observed in complex with the muscarinic M2 [8] or with β1-adrenergic [9] receptor (Figure 5). Interestingly, even though the orientation of bound arrestin relative to the receptor in the case of rhodopsin [5,6], M2 receptor [8], β1-adrenergic receptor [9], vasopressin 2 [62], corticotropin-releasing factor receptor 1 [62], and parathyroid hormone receptor 1 [62], on the one hand, and neurotensin receptor [7,10] on the other differs by almost 90°, the finger loop is invariably found within this receptor cavity. It is worth noting that in the arrestin-3 trimer, where all three protomers were in a receptor bound-like conformation, the finger loops of the three protomers engage each other in a manner mimicking their interaction within the inter-helical cavity of GPCRs [38].

Comprehensive alanine-scanning mutagenesis showed that several mutations in the finger loop affect the receptor binding of arrestin-1 [23,44]. Arrestin-3 is the non-visual subtype that interacts with the widest variety of GPCRs [27,63]. Mutations in its finger loop impacted the interactions with receptors, suggesting that it is an essential element in binding. Deletion of the first glycine in the arrestin-3 finger loop drastically reduced binding to several non-visual GPCRs [64]. Proline substitutions that reduce the flexibility and disrupt the helicity of the finger loop also lowered arrestin-3 binding to tested receptors [38]. Consistent with the direct involvement of this element in receptor binding, numerous finger loop residues were found to be strongly immobilized by the bound receptor in arrestin-1 and arrestin-2 [65,66]. Thus, both structural and biochemical evidence is consistent with the idea that the finger loop acts as a sensor of receptor activation.

The finger loop between β-strands V and VI encompasses 11 residues (68–78) in bovine arrestin-1 [24]. It has the same length in bovine arrestin-2 (64–74) [25] and -3 (65–75) [27]. The role of the finger loop in activation recognition was investigated thoroughly by mutagenesis in arrestin-1 [36]. Finger loop mutations were introduced on the background of WT arrestin-1 and truncated arrestin-1-(1-378), which demonstrates much higher binding to Rh* and inactive P-Rh. The binding to Rh*, which lacks receptor-attached phosphates, is mediated only by the activation recognition, so the finger loop substitutions were expected to impact the binding to Rh* more significantly than to P-Rh*. In contrast, dark P-Rh is inactive, so mutations of residues responsible for the activation recognition were expected to minimally affect the binding to this form of rhodopsin.

Both WT and truncated arrestin-1 exhibited the highest binding to P-Rh* [36]. To specifically study the effects on the binding to the active unphosphorylated receptor, the ability of (1–378) mutant to bind to Rh* at relatively high levels was utilized. The binding of truncated arrestin-1 to Rh* was more sensitive to finger loop substitutions than the binding to dark P-Rh and P-Rh*, consistent with the idea this element is an important site for the recognition of the active receptor conformation. Four key residues were identified (Gly-68, Glu-70, and Ile-72 in the finger loop and immediately following Phe-79 in bovine arrestin-1) (Figure 5) for which mutations had significantly reduced the binding to Rh* but not to other functional forms of rhodopsin [36]. Interestingly, these four residues are conserved in non-visual subtypes [15,67] (Figure 5). The mutagenesis data indicate that these four residues are critical for the activation recognition. This idea is consistent with observed crosslinking of arrestin-2 finger loop residues (Asp-67, Leu-68, Val-70, and Leu-71) with the NTSR1 receptor [7].

All evidence points to the model where the simultaneous engagement of the finger loop and the phosphate sensor by the active phosphorylated receptor allows for arrestin transition from its basal to active state. A distinguishing feature of this transition is the rotation of two domains relative to each other [12]. The finger loop connects β-strands V and VI, which form a β-sheet with β-strands VII and VIII in the N-domain [24,25,26,27]. In the basal state of arrestin, this β-sheet interacts with the loop between β-strands XIV and XV in the C-domain and the lariat loop, which supplies two out of three negative charges of the polar core [24,25,26,27]. It is tempting to speculate that upon receptor binding, the finger loop moves out of its basal position, shifting the four-stranded β-sheet, thus disrupting interactions between the two arrestin domains and facilitating their rotation relative to each other. Molecular dynamics simulations support the idea that receptor interaction with the finger loop increases the likelihood of arrestin domain rotation [28].

## 4. Additional Receptor-Binding Elements

### 4.1. Middle (139) Loop

While the phosphate and activation sensors regulate arrestin activation, structures reveal other elements that contact bound receptor. A strikingly large movement of the loop containing residues 139 and 136 in arrestin-1 and -2, respectively, was observed upon P-Rh* binding. [51,68] (Figure 6). This element is located in the central crest of the receptor-binding surface, next to the finger loop [5]. The large conformation change in this loop was proposed to facilitate binding by making the finger loop and adjacent elements more accessible [68]. This element, remarkably conserved in the arrestin family [15,67], was shown to enhance the stability and selectivity of arrestin-1 [51], as well as arrestin-2 and -3 [68].

Intra-molecular distance measurements in free and P-Rh*-bound arrestin-1 revealed that the 139 loop moves by ~17 Å [51]. The distance changes are consistent with the movement away from the finger loop, which is at the receptor-binding interface [51]. This large movement was suggested to facilitate receptor binding to the finger loop and the other adjacent elements, including the loop containing residue 251, which is immobilized by receptor contact in the complex [51]. Homologous 136-loop in arrestin-2 (called the middle loop [54]) also shows large, although less dramatic movement [68]. Upon P-Rh* binding, the 136-loop moved away from the C-domain and toward the N-domain [68], i.e., in the same direction as the 139 loop in arrestin-1 [51]. The middle (136 and 139) loops show remarkably high mobility in both free arrestin and the P-Rh* complex [51,68]. This flexibility in both the basal and active states likely facilitates the binding of non-visual subtypes, arrestin-2 and -3, to hundreds of different GPCRs with fairly low sequence conservation on the cytoplasmic side [12].

The role of the 139-loop in visual arrestin-1 was further tested by mutagenesis. Deletions in this loop increased P-Rh* binding and reduced thermal stability of arrestin-1, enhancing its ability to bind non-preferred functional forms of rhodopsin [51]. Thus, these deletions reduced arrestin-1 selectivity for P-Rh* [51]. Additionally, the 139-loop deletions disrupted the regulation of the C-tail release from β-strand I, which appears to be necessary for arrestin transition to the active state in the process of P-Rh* binding. The C-tail of the 139-loop deletion mutant was released even in the absence of P-Rh* [51]. Overall, the data demonstrated that the 139-loop stabilizes the basal conformation of arrestin-1 and contributes to its selectivity for P-Rh* by preventing P-Rh and Rh* binding [51].

### 4.2. C-Edge Loops

In addition to the finger loop between β-strands V and VI, structural studies have identified a loop in the C-domain (loop XVIII-XIX or loop-344) that determines differential binding to distinct functional forms of the receptor [69]. Conformational changes in the finger loop were directly linked to receptor activation [69], consistent with the idea that it serves as an activation sensor. In contrast, loop-344 was involved in receptor binding in a manner independent of the receptor activation state. Thus, a binding model was proposed in which the finger loop directly engages the activated receptor, whereas loop-344 engages the membrane or neighboring receptor, regardless of the phosphorylation state of the second receptor [69]. This model was supported by the observed role of the arrestin C-domain in phosphorylation-independent low-affinity receptor binding [69]. Molecular dynamics simulation also revealed membrane-touching loops at the distal tip of the C-domain of arrestin that act as membrane anchors, whose engagement is needed to facilitate GPCR binding by arrestins [6].

### 4.3. Other Arrestin-1 Elements

Structures of arrestin-1 and -2 complexes with several GPCRs revealed other arrestin elements that contact receptors and therefore likely play a role in binding. In the crystal structure of rhodopsin bound to arrestin-1, the phosphorylated rhodopsin–arrestin interface is an intermolecular β-sheet with a network of electrostatic interactions formed between the phosphorylated C-terminus of rhodopsin and the N-terminal domain of arrestin [6]. Phosphorylation was observed at rhodopsin C-terminus residues Thr-336 and Ser-338, which in conjunction with Glu-341 created an extensive network of electrostatic interactions with three positively charged pockets in arrestin-1 [6]. These pockets are formed by three groups of basic residues: Lys-16, Arg-19, and Arg-172 (pocket 1); Lys-16, Arg-30, and Lys-301 (pocket 2); and Lys-15 and Lys-111 (pocket 3), which interact with the rhodopsin C-terminus phosphate groups at Thr-336 and Ser-338, and the negatively charged Glu-341, respectively [6]. The structure with improved resolution [6], as compared to the original one [5], revealed the position of the rhodopsin C-terminus in the complex, which was found to serve as part of the receptor–arrestin interface. The C-terminal portion from Lys-339 to Thr-342 of rhodopsin becomes a β-strand that forms an extended intermolecular antiparallel β-sheet with β-strand I from Val-12 to Lys-16 of the N-terminal domain of arrestin-1 [6]. This rhodopsin β-strand had the same structure in all four complexes in the asymmetric unit, supporting its role as an important part of the arrestin–rhodopsin interface.

Rhodopsin residues Asp-330 through Ser-338 also create an extended stretch, which interacts with arrestin-1 [6]. The rhodopsin region from Asp-330 to Ser-335 is likely dynamic, as these residues slightly vary among the four complexes in the asymmetric unit [6]. In this region, Asp-330 to Ala-333 form a β-turn alongside β-strands I and II of arrestin-1, and negatively charged residues Asp-330 and Glu-332, and the possibly phosphorylated Ser-334 of rhodopsin electrostatically interact with the positively charged patch containing Arg-19, Lys-167, and Lys-168 on the arrestin N-domain [6]. The overall electrostatic balance appears to be maintained in the arrestin–rhodopsin interface, with 9 positive charges on the arrestin-1 N-domain and 8 to 10 negative charges on the interacting rhodopsin element.

### 4.4. Other Arrestin-2 Residues

Several elements in arrestin-2 (i.e., β-arrestin 1) were found in contact with bound GPCRs. In complex with β1-adrenoceptor (β_1_AR), arrestin-2 has the same orientation relative to the receptor as rhodopsin-bound arrestin-1 [9]. However, β_1_AR has a slightly smaller surface area in contact with arrestin, as compared to rhodopsin [9]. The regions of β_1_AR and rhodopsin that contact arrestin-2 and arrestin-1, respectively, are largely conserved in both receptors [9]. Receptor-binding arrestin elements are also similar. The most significant difference is in the finger loop that adopts different conformations in the two arrestin subtypes. The finger loop of arrestin-1 forms a short α-helix within the inter-helical cavity of the receptor, while the finger loop of arrestin-2 forms β-hairpin [9]. The β-hairpin of arrestin-2 inserts deeper into the receptor cavity than the α-helix of arrestin-1 by ~5 Å [9]. Conceivably, the leucine to cysteine change at position 68 of the finger loop in arrestin-2 contributes to this effect [9]. The relative orientation of the two proteins in the muscarinic M2 receptor-arrestin-2 complex [8] is similar to arrestin-2-β_1_AR [9] and arrestin-1-rhodopsin [5,6] complexes.

The arrestin-2–β1AR interface includes 27 residues on the β_1_AR side and 20 residues on the arrestin-2 side that make contacts [9]. Seven structural elements of arrestin-2 contact bound β_1_AR, including the β-strands V, VI, XV, XVII, and XVIII, along with the loops between β-strands V and VI (finger loop), VIII and IX (middle loop), and XV and XVI (C-loop) [9]. The finger loop makes the largest contact with the receptor (37% of the contact surface) [9]. Acidic phospholipids are known to play a significant role in the binding of arrestins to GPCRs [70]. In the arrestin-2-β_1_AR complex, 32 residues were identified that potentially engage lipid headgroups [9]. This suggests an interesting area for future research: testing whether the membrane lipid composition affects the signaling preferences of GPCRs (e.g., signaling bias, reviewed in detail in [71]).

As far as the muscarinic M2 receptor (M2R) is concerned, it should be noted that the arrestin-2 structure was not obtained with the M2R, which has a large third cytoplasmic loop housing all GRK phosphorylation sites [72,73]. This loop was deleted; instead, the phosphorylated vasopressin receptor C-terminus (V2Rpp) was attached to M2R by sortase [8]. The complex of arrestin-2 with this mutant M2R revealed three main interfaces involving the phosphorylated receptor C-terminus, the 7TM core, and intracellular loop 2 on the receptor side [8]. The phosphopeptide of this chimeric M2R binds to a positively charged groove on the N-domain of arrestin-2 by displacing its C-terminus, and likely destabilizing the arrestin polar core, so that the gate loop can move toward the N-domain [8]. A key phosphorylated residue on the C-terminus, Thr-360 in attached V2Rpp, forms interactions with arrestin-2 residues Arg-25 of the N-domain and Lys-294 of the gate loop, stabilizing an active conformation with characteristic interdomain twist [8]. The second interface between the arrestin-2 finger loop and the 7TM bundle involves both electrostatic and hydrophobic interactions [8]. The C-terminal region of the finger loop with residues L68, V70, L71, and F75 participates in hydrophobic interactions with the receptor cavity formed by TMs 3, 5, 6, and intracellular loop 2 [8].

Interestingly, in this structure, the arrestin-2 C-domain edge (residues 191–196) interacts with the membrane and a second loop (residues 330–340) buries itself in the lipid bilayer [8], similar to the proposed interaction of arrestin-1 C-edge with the membrane [74]. Based on this, it was suggested the C-edge–lipid interaction has functional effects, stabilizing the active arrestin-2 conformation [8]. In this work, arrestin binding to GPCRs was hypothesized to be a two-step process, where binding to receptor phosphorylated C-terminus induces arrestin-2 conformational changes that facilitate its coupling to the receptor 7TM bundle.

The arrestin-2–neurotensin receptor 1 (NTSR1) complex involves intermolecular interactions that include the core interface with two separated patches between the central crest loops of arrestin-2 and the intracellular side of the receptor transmembrane domain, and a tail interface between the arrestin-2 N-domain and the receptor C-terminus [7]. One patch of the core interface is the finger loop. The finger loop makes a direct contact with the turn between TM7 and helix 8 of the receptor [7]. In the arrestin-2-NTSR1 complex, the receptor interacts with arrestin-2 via part of its C-terminus, the transmembrane core, and the C-terminal end of the intracellular loop 3 [10]. Importantly, the C-edge, portions of the 340-loop (residues 330–340) and 191-loop (residues 186–198), seem to be in contact with the detergent micelle. In arrestin-1, homologous 344-loop and 197-loop engage the membrane [74].

Intriguingly, β_1_AR [9] and M2R [8] have the same orientation relative to arrestin-2 as rhodopsin relative to arrestin-1 [5,6], whereas NTSR1 is oriented relative to arrestin-2 in a strikingly different manner, rotated by ~90 degrees in both solved structures [7,10]. The finding that the same arrestin can engage different receptors in a distinct fashion suggests that more than one “flavor” of arrestin complex with the same receptor might be possible. In fact, multiple distances were measured by the pulse EPR technique double electron-electron resonance between selected points in rhodopsin and arrestin-1 [5,6]. While the most populated distances matched the structure, the presence of others suggested that different complexes are formed, only one of which is resolved in crystal. The popular hypothesis that the pattern of receptor phosphorylation determines the functional outcome of arrestin binding (barcode hypothesis [75,76]) also implies that arrestin bound to the same differentially phosphorylated receptor can have different conformations.

Recent structural and mutagenesis work identified both phosphate and activation sensors, as well as other receptor-binding residues in arrestins. However, the structures of free arrestins [24,25,26,27,41] show the starting point, while the structures of arrestins bound to GPCRs [5,6,7,8,9,10] or other activators [38,54,55] show the end point. Yet, we still need to understand the molecular mechanism of arrestin transition from one conformation to another. This is critical for GPCR biology and likely for biased signaling. This might have wider implications, as it appears likely that binding-induced conformational rearrangements govern all protein–protein interactions.

## Figures and Tables

**Figure 1 ijms-22-12481-f001:**
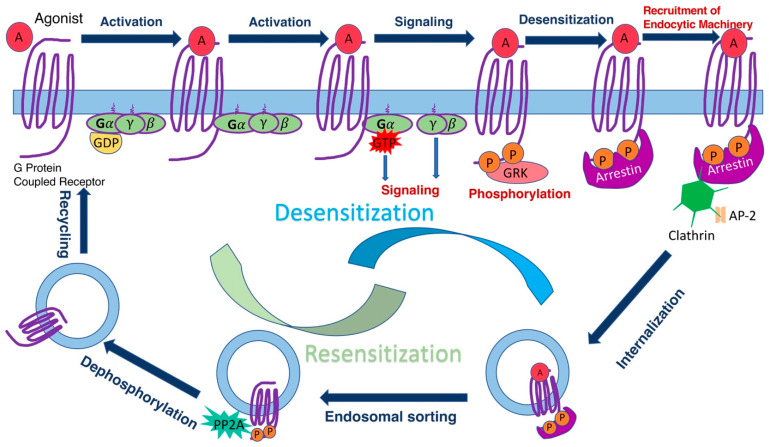
GPCR signaling, desensitization, and internalization. GPCRs (purple) are transmembrane proteins that mostly localize to the plasma membrane (light blue). GPCRs are activated by agonist (A in red circle) binding. Active receptor binds inactive (GDP-liganded) heterotrimeric G protein consisting of α-, β-, and γ-subunits (where α- and γ-subunits have lipid modifications shown as purple tails inserted into the membrane). Receptor binding opens the nucleotide pocket in the G protein α-subunit, leading to the release of bound GDP and binding of GTP, which is more abundant in the cytoplasm. GTP-liganded G protein dissociates from the receptor as a separated α-subunit-GTP and βγ-dimer, both of which interact with various effectors to initiate signaling. Active GPCRs are recognized by specialized GPCR kinases (GRKs, pink oval) that phosphorylate many GPCRs at the C-terminus, although in some receptors, GRK phosphorylation sites are localized on other cytoplasmic elements. Arrestins (magenta) bind active phosphorylated GPCRs and via direct interactions recruit main components of the endocytic machinery of the coated pit, clathrin (green hexagon), and clathrin adaptor AP-2 (yellow), thereby promoting receptor internalization. The internalized receptor is deactivated by the loss of agonist in the acidic environment of the endosome. This facilitates arrestin dissociation, which makes receptor-attached phosphates accessible for the phosphatases (possibly PP2A, shown in green). The dephosphorylated receptor can be recycled back to the plasma membrane and reused. Some internalized GPCRs are sent to lysosomes for degradation (not shown). This reduces the number of receptor molecules in the cell (downregulation).

**Figure 2 ijms-22-12481-f002:**
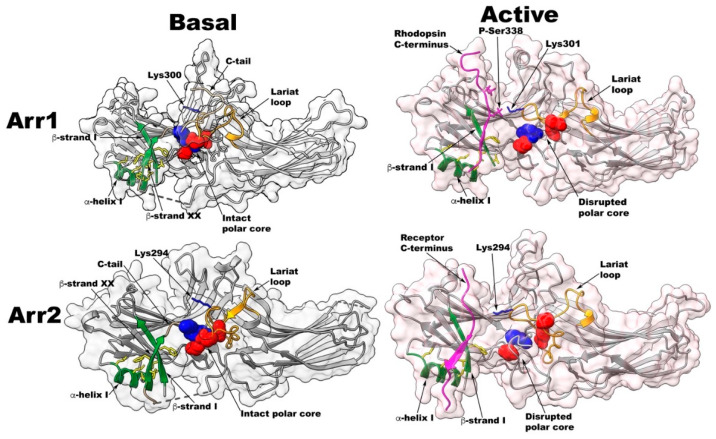
Receptor binding-induced conformational changes in arrestins. Top left. Basal structure of bovine arrestin-1 (PDB ID: 1CF4 [24]). Polar core residues are shown as CPK models (Asp30, Asp296, and Asp303 in red; Arg175 and Arg382 in blue). Lariat loop (containing Asp296, Lys300, and Asp303) is highlighted in yellow. Lys300 is shown as a blue stick model. In the three-element interaction, β-strand I and α-helix I of the N-domain and β-strand XX of the C-tail are shown in green, and bulky hydrophobic residues mediating the interaction are shown as yellow stick models. The part of the arrestin C-tail resolved in the structure is shown in light brown. Missing in structure connection between the C-domain and β-strand XX is shown as a dotted line. Top right. Structure of mouse arrestin-1 bound to human rhodopsin (PDB ID: 5W0P [6]). Note that because of an extra residue at the N-terminus, mouse residue numbers are greater than homologous bovine numbers by one. The C-terminus of bound rhodopsin is shown in magenta. Phosphate attached to rhodopsin Ser338 contacts lariat loop Lys301. Both the polar core and the three-element interaction are disrupted: in the polar core, Asp297 and Asp304 move away from Arg176 and Asp31, Arg residue supplied by the C-tail is absent; in the three-element interaction, β-strand XX of the C-tail is absent. Arrestin elements are shown, as in the top left panel. Lower left. Basal structure of bovine arrestin-2 (PDB ID: 1G4M [25]). Polar core residues are shown as CPK models (Asp26, Asp290, and Asp297 in red; Arg169 and Arg393 in blue). Lariat loop (containing Asp290, Lys294, and Asp297) is shown in yellow. Lys294 is shown as a blue stick model. In the three-element interaction, β-strand I and α-helix I of the N-domain and β-strand XX of the C-tail are shown in green, and bulky hydrophobic residues mediating the interaction are shown as yellow stick models. The part of the arrestin C-tail resolved in the structure is shown in light brown. Missing in the structure connection between the C-domain and β-strand XX is shown as a dotted line. Lower right. Structure of rat cysteine-less arrestin-2-(1–393) bound to human M2 containing the phosphorylated C-terminus of the vasopressin V2 receptor (PDB ID: 6U1N [8]). Residue numbers in rat and bovine arrestin-2 are the same. The V2 C-terminus of the bound receptor is shown in magenta. Attached phosphates were not resolved, but Lys294 points towards the receptor C-terminus. Both the polar core and three-element interaction are disrupted: in the polar core, Asp297 and Asp304 move away from Arg176 and Asp31, Arg residue supplied by the C-tail is absent; in the three-element interaction, β-strand XX of the C-tail is absent. Arrestin elements are shown, as in top left panel. Arr1—arrestin-1, Arr2—arrestin-2. The structures of receptor-bound arrestins have a pinkish tinge.

**Figure 3 ijms-22-12481-f003:**
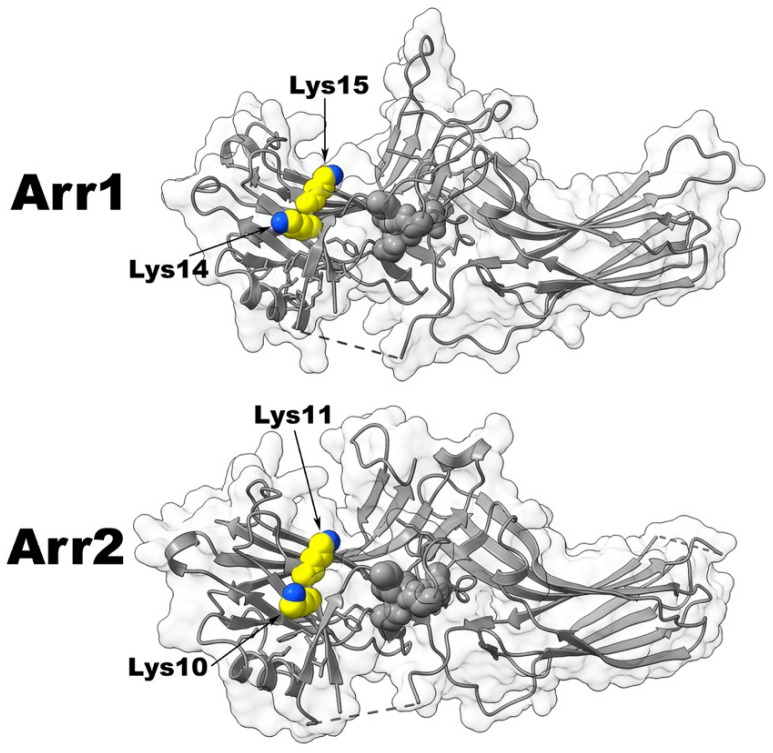
Two lysines in β-strand I of arrestins. Upper panel. The structure of basal bovine arrestin-1 (PDB ID: 1CF4 [24]) with the two β-strand I lysines shown as CPK models. Nitrogen in the amino group is shown in blue, carbon atoms in yellow. Lower panel. The structure of basal bovine arrestin-2 (PDB ID: 1G4M [25]) with the two β-strand I lysines shown as CPK models. Nitrogen in the amino group is shown in blue, carbon atoms in yellow. Arr1—arrestin-1, Arr2—arrestin-2.

**Figure 4 ijms-22-12481-f004:**
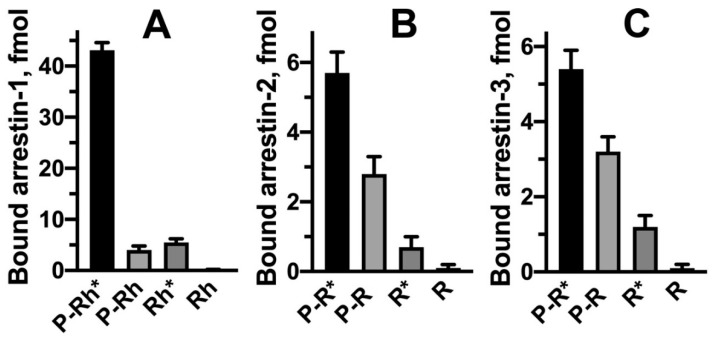
Receptor functional form selectivity of arrestins. (**A**) Binding of arrestin-1 to the four functional forms of rhodopsin. Note the ~10-fold higher binding to P-Rh* than to inactive P-Rh or unphosphorylated active Rh*. (**B**,**C**). Binding of arrestin-2 (**B**) and arrestin-3 (**C**) to four functional forms of the β2-adrenergic receptor. Note only an ~2-fold binding differential between the active and inactive phosphorylated receptor. In all cases, arrestins do not bind inactive unphosphorylated receptors. Binding data from [49].

**Figure 5 ijms-22-12481-f005:**
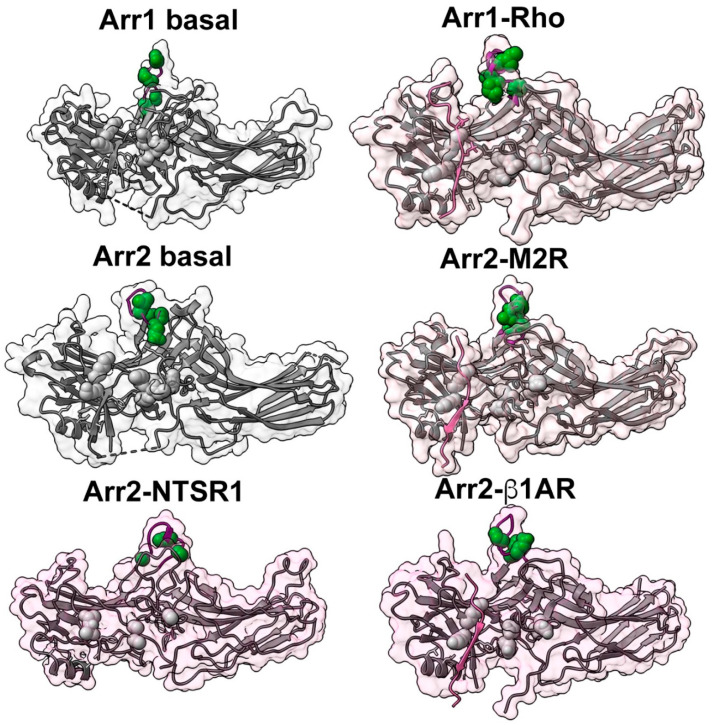
Finger loop as the activation sensor. The structure of the finger loop in basal bovine arrestin-1 (Arr1 basal; PDB 1CF4 [24]), mouse arrestin-1 in complex with rhodopsin (Arr1-Rho, PDB 5W0P [6]), basal arrestin-2 (Arr2 basal, PDB 1G4M [25]), arrestin-2 in complex with M2 muscarinic (Arr2-M2R; PDB 6U1N [8]), β1-adrenergic (Arr2-β1AR; PDB 6TK0 [9]), and NTSR1 neurotensin (Arr2-NTSR1; PDB 6UP7 [10]) receptors. Note that in receptor-bound arrestins, the finger loop can form a short α-helix (Arr1-Rho, Arr2-NTSR1) or a different secondary structure (Arr2-β1AR, Arr2-M2R). Finger loop and adjacent residues identified as critical for recognizing the active state of the receptor in bovine arrestin-1 (Gly68, Glu70, Ile72, Phe79 [36]) and homologous conserved arrestin-2 residues (Gly64, Glu66, Ile68, Phe75) are shown as CPK models in dark green. Polar core residues (Figure 1) and the two lysines in the β-strand I (Figure 2) are shown as grey CPK models. Where resolved in structures, the phosphorylated receptor C-terminus is shown in red. Receptor-bound structures have a pinkish tinge.

**Figure 6 ijms-22-12481-f006:**
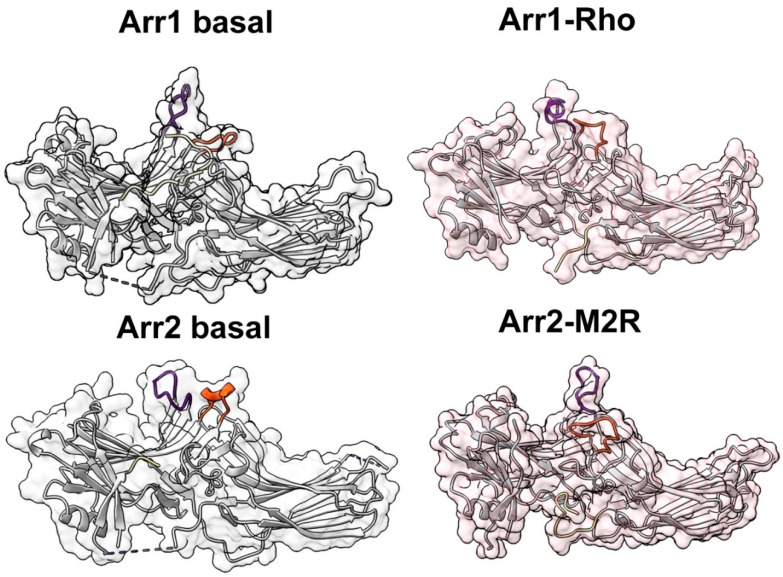
Middle (139) loop. Middle loop in arrestin-2 and homologous 139-loop in arrestin-1 in basal conformation (PDB 1G4M [25] and 1CF4 [24], respectively) and in arrestin-1-rhodopsin (PDB 5W0P [6]) and arrestin-2-M2R (PDB [8]) complexes are shown in light brown. Adjacent finger loops are shown in dark blue. The part of the C-tail not resolved in basal structures is shown as a dotted line. Receptor-bound structures have a pinkish tinge.

**Table 1 ijms-22-12481-t001:** Solved crystal structures of activated arrestins.

Arrestin-Receptor Complex	Crystallography or EM	Protein Modification	Reference	Accession Number
Arrestin-1 and rhodopsin	Serial femtosecond X-ray laser crystallography	Researchers have fused a cysteine-free T4L (residues 2–161 with C54T and C97A) to the N terminus of a rhodopsin that contains four mutations: N2^Nterm^C and N282^ECL3^C to form a disulfide bond, and E113^3.28^Q and M257^6.40^Y for constitutive receptor activity. The C terminus of rhodopsin was fused to 3A_arrestin (L374A, V375A, F376A, residues 10–392) with a 15 amino acid linker (AAAGSAGSAGSAGSA).	Kang, Y., et al., Crystal structure of rhodopsin bound to arrestin determined by femtosecond X-ray laser. Nature, 2015. 523(7562): p. 561–7.	PMID: 26200343 PBD ID: 5W0P
Arrestin-2 and neurotensin	Cryo-electron microscopy structure	Researchers used full-length, native NTSR1 bound to the agonist NTS_8–13_ (amino acids 8–13 of NTS), and phosphorylated the receptor in vitro by G protein coupled receptor kinase subtype 5 (GRK5) using a protocol established for the β_2_ adrenergic receptor.	Huang, W., et al., Structure of the neurotensin receptor 1 in complex with β-arrestin 1. Nature, 2020. 579(7798): p. 303–308.	PMID: 31945771 PBD ID: 6UP7
Pre-activated human arrestin-2-Arg169Glu mutant and β1-adrenergic	Cryo-electron microscopy	Researchers used β_1_AR construct that contained six mutations to improve thermostability and three additional mutations to improve folding and remove palmitoylation. A chimaera between this receptor and the vasopressin V_2_R C terminus enabling efficient in vivo phosphorylation of the receptor and arrestin recruitment was constructed	Lee, Y., et al., Molecular basis of β-arrestin coupling to formoterol-bound β(1)-adrenoceptor. Nature, 2020. 583(7818): p. 862–866.	PMID: 32555462 PBD ID: 6TKO
Truncated arrestin-2 and neurotensin	Cryo-electron microscopy	Researchers fused the wild type human NTSR1 with the human 3A mutant Arr2 at its C-terminus with a three amino acid linker (GSA). Cytochrome b562 RIL domain (BRIL) was fused to the N-terminus of the receptor to increase the complex expression. Arr2 was further stabilized by fusing Fab30 light chain, an antibody fragment used to stabilize the active form of Arr2,^23^ at its C-terminus with a 12 amino acid linker (GSAGSAGSAGSA).	Yin, W., et al., A complex structure of arrestin-2 bound to a G protein-coupled receptor. Cell Res, 2019. 29(12): p. 971–983.	PMID: 31776446 PBD ID: 6PWC
Truncated arrestin-2 and M2 muscarinic cholinergic	Cryo-electron microscopy	Researchers used sortase to enzymatically ligate a synthetic phosphopeptide (pp) derived from the vasopressin-2-receptor (V2R) C-terminus onto the M2R C-terminus (M2Rpp). To enhance arrestin-2 (βarr1) stability, they generated a minimal cysteine variant truncated at residue 393 (βarr1-MC-393)	Staus, D.P., et al., Structure of the M2 muscarinic receptor-β-arrestin complex in a lipid nanodisc. Nature, 2020. 579(7798): p. 297–302.	PMID: 31945772 PBD ID: 6U1N
Arrestin-2 with multi-phosphorylated C-terminal peptide of human V2 vasopressin	Crystallography	Researchers used a conformationally-selective synthetic antibody fragment (Fab30) that recognizes the phosphopeptide-activated state of of arrestin-2 (β-arrestin1).	Shukla, A.K., et al., Structure of active beta-arrestin-1 bound to a G-protein-coupled receptor phosphopeptide. Nature, 2013. 497(7447): p. 137–41.	PMID: 23604254 PBD ID: 4JQI
Arrestin-3 with phosphorylated CXCR7 C-terminal peptide	X-ray crystallography	Researchers used a truncated version of arrestin-3 (βarr2) that lacked the carboxyl-terminal residues 357–410 to facilitate its crystallization in an active conformation, while all other biochemical experiments were performed using full-length arrestin-3 (βarr2).	Min, K., et al., Crystal Structure of beta-Arrestin 2 in Complex wsssith CXCR7 Phosphopeptide. Structure, 2020. 28(9): p. 1014–1023.e4.	PMID: 32579945 PBD ID: 6K3F

## Data Availability

Not applicable.

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
