# Peer review of "Structural Basis of Arrestin Selectivity for Active Phosphorylated G Protein-Coupled Receptors"

_ijms, 2021, doi:10.3390/ijms222212481_

Round 1

Reviewer 1 Report

This is an exciting, well-written, and timely review by a group leader in the field. The same group has published several interesting, somehow overlapping reviews. Still, one of the central aspects of this review (i.e., the differential interaction of arrestin-1 with the different forms of rhodopsin) is exciting to this reviewer. As such, I think the review is excellent but would like the authors to consider the following aspects:

  1. a) Possibly, the title should indicate that most structural data were obtained using the different forms of rhodopsin and their sites of interaction with arrestin-1 (references 17 and 35, among others). There are some data with non-visual arrestins and non-visual GPCR, but the structural information is limited.
  2. b) I would appreciate (and I think other readers would also do so) information on the physical evidence of the four functional forms in non-visual GPCR. Conservation of the finger loop is compelling evidence, but the GPCR family represents around 800 distinct receptors, and variability is likely to occur.
  3. c) Data on GPCR phosphorylation is readily available and also evidence for the phosphate sensor, but that on the GPCR sites that determine activation is not clear to me (please accept my apologies if this is due to my ignorance), and explanation and references would be beneficial and increase the spectrum of interest.
  4. c) Figure 1 is too big, and some aspects were not observable in my pdf; this should be fixed. It would be of interest, if it is possible, to incorporate the different forms of the GPCR in the Figure.
  5. d) For non-visual GPCR, there seem to exist at least two types of interactions with beta-arrestins, one that is relatively transient and another that is much more stable or sustained. Comments on this aspect by the authors would be greatly appreciated.

Minor points

  1. a) In the pdf that I received for evaluation, many "beta" (β) symbols are missing. Similarly, β1-adrenergic receptors are not referred to consistently and should be checked.
  2. b) The Introduction starts with a dot symbol, and in the initiation of some paragraphs, the indentation is missing.

Author Response

This is an exciting, well-written, and timely review by a group leader in the field. The same group has published several interesting, somehow overlapping reviews. Still, one of the central aspects of this review (i.e., the differential interaction of arrestin-1 with the different forms of rhodopsin) is exciting to this reviewer. As such, I think the review is excellent but would like the authors to consider the following aspects:

Thank you!

  1. a) Possibly, the title should indicate that most structural data were obtained using the different forms of rhodopsin and their sites of interaction with arrestin-1 (references 17 and 35, among others). There are some data with non-visual arrestins and non-visual GPCR, but the structural information is limited.

The reviewer is correct, most of the info comes from arrestin-1-rhodopsin data. However, the thrust of the review is extrapolating this info to other arrestins. Therefore, we’d prefer to keep the title as it is. To acknowledge this fact, we added a sentence to the abstract stating that most data were obtained with the arrestin-1-rhodopsin pair.

  1. b) I would appreciate (and I think other readers would also do so) information on the physical evidence of the four functional forms in non-visual GPCR. Conservation of the finger loop is compelling evidence, but the GPCR family represents around 800 distinct receptors, and variability is likely to occur.

Thanks! Done.

  1. c) Data on GPCR phosphorylation is readily available and also evidence for the phosphate sensor, but that on the GPCR sites that determine activation is not clear to me (please accept my apologies if this is due to my ignorance), and explanation and references would be beneficial and increase the spectrum of interest.

Thanks! Done.

  1. c) Figure 1 is too big, and some aspects were not observable in my pdf; this should be fixed. It would be of interest, if it is possible, to incorporate the different forms of the GPCR in the Figure.

Thanks! The figure was modified.

  1. d) For non-visual GPCR, there seem to exist at least two types of interactions with beta-arrestins, one that is relatively transient and another that is much more stable or sustained. Comments on this aspect by the authors would be greatly appreciated.

Thanks for pointing this out. We mentioned this possibility. Even though this was demonstrated using an artificial construct, b2-adrenergic receptor with vasopressin V2 receptor C-terminus, it is entirely possible that this might be applicable to natural non-visual GPCRs.

Minor points

  1. a) In the pdf that I received for evaluation, many "beta" (β) symbols are missing. Similarly, β1-adrenergic receptors are not referred to consistently and should be checked.

Thanks! Corrected.

  1. b) The Introduction starts with a dot symbol, and in the initiation of some paragraphs, the indentation is missing.

Thanks! Corrected.

Reviewer 2 Report

The manuscript written by Preethi C. Karnam et al. entitled “Structural basis of arrestin selectivity for active phosphorylated G protein-coupled receptors” is a short and focused review on how arrestins achieve selectivity for the active conformation and the phosphorylated receptor state. This review is written by one of the leading groups in this field and will help new researchers to get started. Nonetheless, I have the feeling this manuscript was written quit fast and would benefit from a bit of editing work as outlined in the following points of critzicsm.  

General points.

1) Within the entire manuscript the authors need to add the greek letters alpha and beta to all alpha helices and beta sheets that are described throughout the manuscript. In the pdf I had to review there were none which might be due to the use of a wrong font. Please correct this.

2) The authors describe a table 1 but this was not included within the manuscript. This must be provided.

3) Figure 1 is partly cut and not fully visible. This need to be corrected as well.

4) The authors are, of course, very familiar with arrestin-specific terminology. For the more general readership, the utilization of the word “sensor” might lead to some confusion. Especially, in the beginning of the manuscript, “sensor” is used in a rather unspecific manner. The authors should make it clear from the beginning that the word “sensor” refers to functional domains or residues inside the arrestin protein, which are important for the recognition of active receptor folds or phosphorylated peptide stretches.

5) The discussed data on arrestin mutations is accurately cited. Nevertheless, the review should at least mention and acknowledge the existence of two publications, which constitute the comprehensive alanine scan of arrestin-1.

https://doi.org/10.1073/pnas.1319402111 and

https://doi.org/10.1038/srep30224.

6) Regarding figures 2, 3, 5, and 6. The authors should consider using (PDB: 7JSM) instead of (PDB: 1CF4) since this very recently published structure features most of the arrestin-1 C-terminal tail. This might also solve a related problem: in line 99 and 113, the authors claim that the resolved part of the arrestin C-terminus is colored in light brown. I was not able to spot this in the figure (which might be due to the lower resolution of the provided pdf) and, in any case, only a small part of the arrestin-1 C-tail is resolved in the original 1CF4 structure.

Minor comments:

Line 9-15        The authors should consider to specify some aspects of their abstract for the broad readership. First, arrestin is binding to active phosphorylated GPCRs with higher affinity than to other forms. The phrasing “arrestin binding … is much greater…” (line 9) might not be as clear. Second, the authors state that arrestins “must have separate sensors” (line 11) and go on to describe that the “engagement of both sensors enables arrestin transition” (line 12). For readers without in depth knowledge of this field it might be unclear that this directly refers to “detecting receptor-attached phosphates” and “active receptor confirmation” separately. Additionally, the phrasing “brings additional elements of the arrestin molecule in contact with a GPCR” (line 14) is very vague and might benefit from a specific example. In line 14-15, the sentence should read “…, thereby stabilizing the complex”.

line 24, “even more in most mammalian species” needs a reference.

line 39, “GPCRs exist in the plasma membrane” should be “GPCRs are transmembrane proteins that mostly localize to the plasma membrane”.

lines 64, 460, “visual arrestin-1” seems redundant.

line 65, “phosphoreceptor” might be better called “phosphorylated photoreceptors”?

lines 68-71, (https://doi.org/10.1038/s41598-018-36881-4) shows that arrestin-1 also has an affinity towards phosphorylated opsin, this receptor state should be mentioned. Additionally, the authors state multiple times that rhodopsin exhibits “four functional” forms (e.g. line 286). This is a rather inaccurate depiction of the different activation states of rhodopsin, especially considering that 11-cis retinal acts as an inverse agonist.

Line 76, 77      The sentence should rather read “engage active receptor and phosphorylated GPCR C-terminus” instead of “…and phosphate sensors”. In the beginning, the authors refer to the arrestin with “phosphate sensor” but here they seem to refer to the interaction sites on the receptor, which could lead to confusion.

Line 79, 80      “that increase the energy of interaction” could more correctly be expressed as  “that stabilize this interaction, and therefore increase the binding affinity”.

line 82, “the exact mechanism of this conformational change…” Although this might not be the immediate focus of the review, but recent advances the assessment of arrestin conformational changes should be mentioned here.

https://doi.org/10.1038/nature17154,

https://doi.org/10.1038/nature17198,

https://doi.org/10.1038/s41467-019-09204-y,

https://doi.org/10.1016/j.cell.2020.11.014.

line 87, “phosphate sensor” should be “phosphate sensors”.

lines 199, 220, 228, 240, the term “pre-activated” is appropriately used. Yet, the authors should consistently use quotation marks or remove them. Furthermore, the review would benefit from a comprehensive definition of “inactive/basal”, “pre-active”, and “active” states of arrestin, especially since these terms are not always used in the same context in scientific literature.

line 219, “phosphor-receptor” should be “phosphorylated receptor”?.

Line 274          The authors might consider to use the term “significantly less” to describe the pronounced difference in selectivity for the P-R* state between arrestin-1 and arrestin-2, -3. This might lead to the expectation of a statistical analysis, which was not indicated here.

Lines 301-303, “side” should be “site”. Additionally, the sentence would benefit from semantic/grammatical restructuring. It currently reads: “In arrestin-1 complex with rhodopsin, the unstructured in the basal state finger loop in the central crest of the receptor-binding side becomes…” The construction of the sentence makes it hard to follow what the authors intend to state.  

Line 351 The authors mention the identification of four key residues in the finger loop region of bovine arrestin-1. For the reader less acquainted with arrestins in general, it would be also a helpful information, which amino acids are considered to be part of the finger loop region and therefore how large this region is in total. Please consider mentioning in the text when introducing the finger loop region.

lines 362-363, “in the basal arrestin” should be “in the basal state of arrestin”.

Line 389, 466                “~17 A” should be “~17 Å” and “~5 A”should be “~5 Å”, respectively.

line 429, “phosphorhodopsin” should be “phosphorylated rhodopsin”.

lines 456-457, “were found in contact the GPCRs it binds” should be “were found in contact with the GPCRs it binds”. Additionally, this sentence seems redundant.

line 470, “and and” should be “and”.

line 486, “fusion” consider using “chimeric”.

Comment to figure 1

Figure 1          The labels for the arrows that should read “desensitization” and “resensitization” could be placed more conveniently, to avoid confusion as it can also be interpreted as the tips of the arrows pointing to the opposite label. In the figure legend there seems to be a problem with symbols like alpha, beta, gamma (which holds true throughout the manuscript as mentioned above). Lipid modifications are shown as purple tails inserted into the membrane, as opposed to black tails as written in the figure legend.

Author Response

The manuscript written by Preethi C. Karnam et al. entitled “Structural basis of arrestin selectivity for active phosphorylated G protein-coupled receptors” is a short and focused review on how arrestins achieve selectivity for the active conformation and the phosphorylated receptor state. This review is written by one of the leading groups in this field and will help new researchers to get started.

 Thanks!

General points.

1) Within the entire manuscript the authors need to add the greek letters alpha and beta to all alpha helices and beta sheets that are described throughout the manuscript. In the pdf I had to review there were none which might be due to the use of a wrong font. Please correct this.

Thanks! Corrected.

2) The authors describe a table 1 but this was not included within the manuscript. This must be provided.

Thanks for pointing out this omission! Table 1 is now included.

3) Figure 1 is partly cut and not fully visible. This need to be corrected as well.

Fig. 1 was modified to eliminate the impression that it is incomplete.

4) The authors are, of course, very familiar with arrestin-specific terminology. For the more general readership, the utilization of the word “sensor” might lead to some confusion. Especially, in the beginning of the manuscript, “sensor” is used in a rather unspecific manner. The authors should make it clear from the beginning that the word “sensor” refers to functional domains or residues inside the arrestin protein, which are important for the recognition of active receptor folds or phosphorylated peptide stretches.

Thank you! A better explanation is presented.

5) The discussed data on arrestin mutations is accurately cited. Nevertheless, the review should at least mention and acknowledge the existence of two publications, which constitute the comprehensive alanine scan of arrestin-1.

https://doi.org/10.1073/pnas.1319402111 and

https://doi.org/10.1038/srep30224.

Thanks! These references were included and the data generated in these studies were discussed.

6) Regarding figures 2, 3, 5, and 6. The authors should consider using (PDB: 7JSM) instead of (PDB: 1CF4) since this very recently published structure features most of the arrestin-1 C-terminal tail. This might also solve a related problem: in line 99 and 113, the authors claim that the resolved part of the arrestin C-terminus is colored in light brown. I was not able to spot this in the figure (which might be due to the lower resolution of the provided pdf) and, in any case, only a small part of the arrestin-1 C-tail is resolved in the original 1CF4 structure.

We thank the reviewer for this suggestion. Indeed, the structure in 7JSM has higher resolution than 1CF4, and shows the position of much greater portion of the arrestin-1 C-terminus. We now cited this study, which reveals possible role of IP6 in arrestin-1 function. However, this structure appeared after all the mutagenesis studies discussed here were done. As these studies, the results of which are illustrated in figures, were based on 1CF4, we believe that using that structure, rather than retroactively fitting the data to 7JSM, is more appropriate. Besides, the parts of arrestin-1 resolved in both structures have essentially the same positions. Thus, the logic based on 1CF4 remains valid after the publication of 7JSM. 

Minor comments:

Line 9-15        The authors should consider to specify some aspects of their abstract for the broad readership. First, arrestin is binding to active phosphorylated GPCRs with higher affinity than to other forms. The phrasing “arrestin binding … is much greater…” (line 9) might not be as clear. Second, the authors state that arrestins “must have separate sensors” (line 11) and go on to describe that the “engagement of both sensors enables arrestin transition” (line 12). For readers without in depth knowledge of this field it might be unclear that this directly refers to “detecting receptor-attached phosphates” and “active receptor confirmation” separately. Additionally, the phrasing “brings additional elements of the arrestin molecule in contact with a GPCR” (line 14) is very vague and might benefit from a specific example. In line 14-15, the sentence should read “…, thereby stabilizing the complex”.

Thanks! That part of the text was extensively modified.

line 24, “even more in most mammalian species” needs a reference.

Thanks. The reference to the GPCR database is added.

line 39, “GPCRs exist in the plasma membrane” should be “GPCRs are transmembrane proteins that mostly localize to the plasma membrane”.

Thanks! Corrected.

lines 64, 460, “visual arrestin-1” seems redundant.

Thanks! Corrected.

line 65, “phosphoreceptor” might be better called “phosphorylated photoreceptors”?

Thanks! Corrected.

lines 68-71, (https://doi.org/10.1038/s41598-018-36881-4) shows that arrestin-1 also has an affinity towards phosphorylated opsin, this receptor state should be mentioned. Additionally, the authors state multiple times that rhodopsin exhibits “four functional” forms (e.g. line 286). This is a rather inaccurate depiction of the different activation states of rhodopsin, especially considering that 11-cis retinal acts as an inverse agonist.

Thanks for pointing this out! The reviewer is correct, and we now included the mention of opsin and phosphorylated opsin as the two additional functional forms of rhodopsin and discuss the data obtained with opsin.

Line 76, 77      The sentence should rather read “engage active receptor and phosphorylated GPCR C-terminus” instead of “…and phosphate sensors”. In the beginning, the authors refer to the arrestin with “phosphate sensor” but here they seem to refer to the interaction sites on the receptor, which could lead to confusion.

Thanks! Rephrased. However, as many GPCRs have GRK phosphorylation sites in other than the C-terminus elements, we did not use the term “C-terminus”.

Line 79, 80      “that increase the energy of interaction” could more correctly be expressed as  “that stabilize this interaction, and therefore increase the binding affinity”.

Thanks! Corrected.

line 82, “the exact mechanism of this conformational change…” Although this might not be the immediate focus of the review, but recent advances the assessment of arrestin conformational changes should be mentioned here.

https://doi.org/10.1038/nature17154,

https://doi.org/10.1038/nature17198,

https://doi.org/10.1038/s41467-019-09204-y,

https://doi.org/10.1016/j.cell.2020.11.014.

Thanks for pointing out these studies! They are now cited and discussed.

line 87, “phosphate sensor” should be “phosphate sensors”.

The model calls for one, although does not exclude more. However, existing mutagenesis data suggest that a single sensor in arrestin “detects” GPCR phosphorylation.

lines 199, 220, 228, 240, the term “pre-activated” is appropriately used. Yet, the authors should consistently use quotation marks or remove them. Furthermore, the review would benefit from a comprehensive definition of “inactive/basal”, “pre-active”, and “active” states of arrestin, especially since these terms are not always used in the same context in scientific literature.

Great suggestion! We used the quotation marks consistently and added the explanation of our usage of these terms (which is, indeed, inconsistent in various studies).

line 219, “phosphor-receptor” should be “phosphorylated receptor”?.

Thanks! Corrected.

Line 274          The authors might consider to use the term “significantly less” to describe the pronounced difference in selectivity for the P-R* state between arrestin-1 and arrestin-2, -3. This might lead to the expectation of a statistical analysis, which was not indicated here.

Thanks! Corrected.

Lines 301-303, “side” should be “site”. Additionally, the sentence would benefit from semantic/grammatical restructuring. It currently reads: “In arrestin-1 complex with rhodopsin, the unstructured in the basal state finger loop in the central crest of the receptor-binding side becomes…” The construction of the sentence makes it hard to follow what the authors intend to state.  

Thanks! Rephrased.

Line 351 The authors mention the identification of four key residues in the finger loop region of bovine arrestin-1. For the reader less acquainted with arrestins in general, it would be also a helpful information, which amino acids are considered to be part of the finger loop region and therefore how large this region is in total. Please consider mentioning in the text when introducing the finger loop region.

 Thanks! Done.

lines 362-363, “in the basal arrestin” should be “in the basal state of arrestin”.

Thanks! Corrected.

Line 389, 466                “~17 A” should be “~17 Å” and “~5 A”should be “~5 Å”, respectively.

Thanks! Corrected.

line 429, “phosphorhodopsin” should be “phosphorylated rhodopsin”.

Thanks! Corrected.

lines 456-457, “were found in contact the GPCRs it binds” should be “were found in contact with the GPCRs it binds”. Additionally, this sentence seems redundant.

Thanks! Corrected.

line 470, “and and” should be “and”.

Thanks! Corrected.

line 486, “fusion” consider using “chimeric”.

Thanks! Rephrased.

Comment to figure 1

Figure 1          The labels for the arrows that should read “desensitization” and “resensitization” could be placed more conveniently, to avoid confusion as it can also be interpreted as the tips of the arrows pointing to the opposite label. In the figure legend there seems to be a problem with symbols like alpha, beta, gamma (which holds true throughout the manuscript as mentioned above). Lipid modifications are shown as purple tails inserted into the membrane, as opposed to black tails as written in the figure legend.

Thanks for noticing! The figure was modified.